# Is the Social Gradient in Net Survival Observed in France the Result of Inequalities in Cancer-Specific Mortality or Inequalities in General Mortality?

**DOI:** 10.3390/cancers15030659

**Published:** 2023-01-20

**Authors:** Laure Tron, Laurent Remontet, Mathieu Fauvernier, Bernard Rachet, Aurélien Belot, Ludivine Launay, Ophélie Merville, Florence Molinié, Olivier Dejardin, Guy Launoy

**Affiliations:** 1ANTICIPE U1086 INSERM-UCN, Equipe Labellisée Ligue Contre le Cancer, Centre François Baclesse, Normandie Université UNICAEN, 14000 Caen, France; 2Service de Biostatistique—Bioinformatique, Pôle Santé Publique, Hospices Civils de Lyon, 69000 Lyon, France; 3University of Lyon, 69000 Lyon, France; 4University of Lyon 1, 69100 Villeurbanne, France; 5Équipe Biostatistique-Santé, Laboratoire de Biométrie et Biologie Évolutive, CNRS, UMR 5558, 69100 Villeurbanne, France; 6Inequalities in Cancer Outcomes Network, Department of Non-Communicable Disease Epidemiology, Faculty of Epidemiology and Population Health, London School of Hygiene and Tropical Medicine, London WC1E 7HT, UK; 7French Network of Cancer Registries (FRANCIM), 31000 Toulouse, France; 8Loire-Atlantique-Vendée Cancer Registry, 44000 Nantes, France; 9Centre d’Epidémiologie et de Recherche en santé des POPulations (CERPOP) UMR1295, Université de Toulouse Paul Sabatier, Inserm, 31000 Toulouse, France; 10Research Department, Caen University Hospital Centre, 14000 Caen, France

**Keywords:** France, cancer registries, excess mortality framework, simulated deprivation-specific life tables, European deprivation index

## Abstract

**Simple Summary:**

That the original French life tables are not stratified in terms of deprivation whilst the background mortality in the general population differs according to socio-economic position, social gradient in the net survival of patients with cancer, as was found in a previous study, could be due, at least partly, to socially-determined co-morbidities. We found that the social gradient in cancer net survival was reduced using simulated deprivation-specific life tables. This study alerts us to the fact of this overestimation in the social gradient in cancer net survival using the original life tables, which, in a few cases, can be so important that conclusions might be wrong (e.g., prostate cancer). As this work relies upon simulated rather than real data, we were not able to precisely quantify the potential bias resulting from the lack of deprivation-specific life tables. This present study points to how important it is to create proper deprivation-specific life tables in order to accurately investigate social inequalities in cancer net survival analyses.

**Abstract:**

Background: In cancer net survival analyses, if life tables (LT) are not stratified based on socio-demographic characteristics, then the social gradient in mortality in the general population is ignored. Consequently, the social gradient estimated on cancer-related excess mortality might be inaccurate. We aimed to evaluate whether the social gradient in cancer net survival observed in France could be attributable to inaccurate LT. Methods: Deprivation-specific LT were simulated, applying the social gradient in the background mortality due to external sources to the original French LT. Cancer registries’ data from a previous French study were re-analyzed using the simulated LT. Deprivation was assessed according to the European Deprivation Index (EDI). Net survival was estimated by the Pohar–Perme method and flexible excess mortality hazard models by using multidimensional penalized splines. Results: A reduction in net survival among patients living in the most-deprived areas was attenuated with simulated LT, but trends in the social gradient remained, except for prostate cancer, for which the social gradient reversed. Flexible modelling additionally showed a loss of effect of EDI upon the excess mortality hazard of esophagus, bladder and kidney cancers in men and bladder cancer in women using simulated LT. Conclusions: For most cancers the results were similar using simulated LT. However, inconsistent results, particularly for prostate cancer, highlight the need for deprivation-specific LT in order to produce accurate results.

## 1. Introduction

Social inequalities in health are a major concern. Whatever the disease, in order to better understand the underlying mechanisms of the social gradient in survival—and to be able to propose targeted actions to remedy it—it is necessary to know to what extent it can be attributed to general mortality or disease-specific mortality. In the cancer field, it is important to make this distinction in order to know to what extent the observed social gradient in survival is due to differences in management (screening, diagnostic, treatment, and follow-up) associated with the social environment.

Social inequalities in cancer survival have been reported worldwide and represent a major public health burden [1,2,3,4,5,6,7,8,9,10]. As a matter of fact, public health policies, both at the national (French cancer plans and Stratégie Décennale Santé) and the European (Cancer Mission) level are focused on the reduction of social inequalities in cancer screening, incidence, and survival.

In France, a recent study [11] investigated the social gradient in cancer survival for 19 major solid tumors (about 200,000 cancer cases) using data from the French Network of Cancer registries (Francim). The study found lower 5-years survival among patients living in the most-deprived areas, as compared to patients living in the least-deprived ones, for almost all cancer sites. This reduction in survival probability among the most deprived reached 60% for bile ducts in women, and 35% for liver cancer in men. In this study, net survival was estimated using the *relative survival setting* or *excess mortality framework* [12]. In this setting, the knowledge of the cause of death of cancer patients is not required, but appropriate life tables (LT) are needed. Indeed, net survival for a given cancer is a conceptual and theoretical measure [13,14] derived from the excess (or cancer-specific) mortality, this latter being the difference between the observed mortality of cancer patients and their expected or background mortality, known from general population LT. However, as in other countries, an important limitation of this previous work was the lack of stratification in terms of deprivation of the French LT produced by the National Institute of Statistics and Economic Studies (INSEE), which prevented accounting for the expected social gradient in general mortality [5,15].

As a matter of fact, mortality in the general population is known to be socially differentiated. Mackenbach and colleagues [16] reported higher rates of mortality among low- versus highly educated people in several European countries. For example, in France, mortality is doubled among the least educated compared with the most educated, and in Lithuania, mortality is up to three times higher among people with low education. The same pattern was recently found in England and Wales [17]. Therefore, in net survival analyses using relative survival setting, if social inequalities in background mortality in the general population are not properly accounted for, they will be included in the social gradient in excess mortality in the cancer registries population: this will mathematically result in an overestimation of the social gradient in excess mortality [18]. In other words, in net survival analyses, without stratification of LT by social situation, if there is a social gradient in the observed mortality, a social gradient in excess mortality (likewise in net survival) will be found, but it might result totally or partly from the unaccounted social gradient in the expected mortality.

As original French LT ignore the social inequalities in mortality in the general population (being stratified only by sex, age, year, and French Département, which is a territorial and administrative division in France) an overestimation of the social gradient in cancer net survival cannot be excluded in this previous French study [11].

Correcting methods for excess mortality models are thus being developed [19,20,21], but to date, they are mainly experimental. One solution is to conduct sensitivity analyses using simulated deprivation-specific LT [5,12,15]. Using this method, Antunes and colleagues [15] have shown that, in Portugal, social inequalities in colorectal cancer net survival were most likely attributable to inequalities in background survival. However, Ito and colleagues [5] still found social inequalities in cancer net survival for several cancers in sensitivity analyses, and showed that the overestimation of the social gradient might be smaller at 1 year than at 5 years of follow-up.

In that context, our aim was to evaluate whether the social deprivation gap in cancer net survival observed in France could be attributable to this lack of appropriate LT. To accomplish this, we simulated deprivation-specific French LT and compared net survival analyses using original LT versus net survival analyses using simulated deprivation-specific LT, thus conducting sensitivity analyses for results from Tron et al. 2019, providing social gradient in survival for almost all cancer sites using French population-based registries data [11].

## 2. Materials and Methods

### 2.1. Cancer Registries Data

Data were collected from French cancer registries (Francim) as described previously [11]. Cancer cases diagnosed between 1 January 2006 and 31 December 2009 in patients over 15 years old were included and followed-up until the date of death or 30 June 2013. The quality of the data derived from the registries was regularly assessed by the National Committee of Registries (CER) at the national level and by the International Agency for Research on Cancer (IARC) at the international level. The study was approved by the Consultative Committee for the Processing of Health Research Data (CCTIRS) and the French Data Protection Authority (CNIL, authorization n°913013). Available variables were survival time (i.e., difference between the date of death or last information on vital status and the date of cancer diagnosis), sex, age at diagnosis, and social environment, assessed by the 2007 version of the European Deprivation Index (EDI) [22,23]. The EDI is an aggregate ecological measure of relative poverty (deprivation), available in France at the “IRIS” (Ilots Regroupés pour l’Information Statistique) level and based on census data and information from the European Union Statistics on Income and Living Conditions survey (EU-SILC). IRIS are small (around 2000 inhabitants) geographic areas created by INSEE, for which census data are available. For each cancer case recorded in the registries, the patient’s address was geolocalized using Geographic Information Systems (ArcGIS 10.5, ESRI, Redlands, CA, USA) in order to be allocated to an IRIS and thus the corresponding EDI score. The EDI score is a continuous variable ranging from −17 to 51 (median value: −0.75) at the national level—the higher the score, the greater the deprivation in the IRIS. It has also been categorized according to quintiles of distribution at the national level (Q1 referring to the least- and Q5 to the most-deprived environment). Missing data for EDI accounted for less than 1%; therefore, we performed complete cases analyses.

### 2.2. Simulation of Deprivation-Specific French LT

Following the same methodology used in previous studies [5,12,15,24,25], two sets of simulated deprivation-specific French LT were created by applying, to the original French LT, the social gradient in mortality observed in two external sources. The first source was the deprivation-specific LT available for the general population of England (Eng LT) [26], which provide the mortality rate ratios from 2006 to 2011, by sex, age, and national quintiles of the income domain score of the Indice of Multiple Deprivation (IMD)—an aggregated ecological index at the contextual level [27]. England deprivation-specific LT were chosen, because they are fully available, and because England is known to have large mortality inequalities, as is France [16]. The second source was a large-scale socio-demographic panel established in France in the general population, the Echantillon Démographique Permanent (EDP, Permanent Demographic Sample) [28] which provides the mortality rate ratios (mortality quotient per 100,000 people) for the period 2012–2016, by sex, age and by educational level, socio-professionnal category or net income per consumption unit of household. We chose to use the net income per consumption unit of household —a single indicator at the individual level for better comparability. In the EDP panel, 20-quantiles of net income per consumption unit of household were available, and we calculated the means of 20-quantile 1 to 4, 5 to 8, 9 to 12, 13 to 16, and 17 to 20 in order to obtain a variable with 5 categories for the analyses.

In both cases (Eng LT and EDP), we extracted the smoothed expected mortality rates from the external source according to five levels of deprivation. Next, we estimated the mortality rate ratios between each of these five categories and the overall general mortality provided by the external source (see Appendix A). Finally, for each stratum of sex, age, year, and French Département in the French LT, we applied these rate ratios to the original mortality rate in order to obtain new ones for 5 levels of deprivation. Thus, we applied the social gradient in mortality observed in the general population from external sources (Eng LT and EDP) to the original French LT. This resulted in simulated French LT stratified by sex, age, year, French Département, and deprivation. The detailed calculation process is presented in Appendix A.

### 2.3. Statistical Analyses

In the relative survival setting, net survival can be studied using non-parametric Pohar–Perme method [14] or by the flexible modelling of excess mortality hazard using multidimensional penalized splines [29,30]. These two methods have been used in the previous studies using the original French LT; results from the Pohar–Perme method have been published in Tron et al., 2019 [11], whereas results from flexible modelling have been published in Poiseuil et al., 2022 [31] and Tron et al., 2021 [32].

In the present study (which corresponds to a sensitivity analysis of the previous ones), we then compared these initial results with those obtained using the same exact methodology (Pohar–Perme and modelling) but using the two sets of simulated LT. We recall below some characteristics concerning the methods.

(i)The 5-year age-standardized net survival probabilities were estimated using the Pohar–Perme method [14]. For each cancer site and sex, we calculated the 5-year deprivation gaps (and their 95% confidence intervals [CI]), which are the difference in 5-year age-standardized net survival probabilities between patients from the least- and most-deprived environments defined by the 1st and 5th national quintile of EDI, respectively (see Tron et al., 2019 [11] method section for further calculation details).(ii)In flexible modelling, at given values of time (t), age at diagnosis, and EDI, the observed mortality hazard λ of a given patient is decomposed as follows: λ(t,age,EDI,z) = λ_E_(t,age,EDI) + λ_P_(age + t,year + t,z), where λ_E_ is its excess mortality hazard (EMH), which is the mortality directly or indirectly due to cancer, and λ_P_ is its expected mortality, i.e., the all-cause mortality hazard of the general French population at age at diagnosis + t, and year of diagnosis + t, given demographic characteristics z of that individual. Here z is composed of the variables sex, year of death, residence French Département, and deprivation in the simulated LT. The EMH was modeled using (multidimensional) penalized splines, which allows for the modelling of flexible baseline hazard, the non-linear and non-proportional (i.e., time-dependent) effects of covariates, as well as interactions [29,30]. More precisely, four models based on penalized splines were adjusted, and the best one was selected according to the corrected Akaike Information Criterion (AIC) [33] indicating the overall effect of EDI on cancer net survival and its form, either 1—no effect; 2—proportional (i.e., not time-dependent) effect; 3—time-dependent effect; or 4—time- and/or age-dependent effect (i.e., interaction EDI*t and/or EDI*age). Then, excess mortality hazard ratios (EHR) by EDI were calculated based on the selected model. See the methods section of Poiseuil et al., 2022 [31] and Tron et al., 2021 [32] for further details about the modelling strategy.

For each cancer and sex, three analyses were compared: main analyses based on the original French LT, sensitivity analyses using English LT, and external source and sensitivity analyses using EDP external source. The following elements were compared between the three analyses:-value of the 5-year deprivation gaps (and 95% CI)-the selected flexible model (indicating the effect of EDI on survival and its form)-the curves of EHR as a function of EDI (using the 10th percentile of EDI as a reference, i.e., EDI score = −3.9)-and the curves of the EHR of the 90th percentile (p90 = 4.4) as compared to the 10th percentile (p10 = −3.9) of EDI as a function of time since cancer diagnosis.

All of the described models were fitted using R software (R Core Team, Vienna, Austria, version 3.5.1) with package ‘relsurv’ (2.2.3) and ‘survPen’ (1.0.1) [34].

### 2.4. Data Availability Statement

The data generated in this study are available upon request from the corresponding author.

## 3. Results

A total of 185,169 solid tumors (18 different cancer sites, 14 in men and 17 in women) were analyzed as in Tron et al., 2019 [11] (see Table 2 of this previous article for the description of the studied population and cancer sites).

### 3.1. Comparison of Main and Sensitivity Analyses Based on Non-Parametric Method

Overall, 5-year deprivation gaps were reduced in both sensitivity analyses as compared to main analyses.

In men (Figure 1a), clear 5-year deprivation gaps were observed in all three analyses for ear, neck, throat (ENT) cancers and liver cancer. The results regarding kidney, stomach, pancreas, central nervous system (CNS), and bile ducts cancers as well as melanoma and sarcomas were identical according to the three analyses, with negligible 5-year deprivation gaps (DG) in any of them. For some cancer sites such as colon–rectum, bladder, and lung, the 5-year deprivation gap became negligible in both sensitivity analyses, but a similar trend remained as compared to main analyses. Inconsistent results were observed for prostate cancer, for which the simulated LT not only reduced the social gradient in net survival but also inversed it (DG_main analyses_: 3.0, 95% CI: 1.1;4.9, DG_SA_Eng LT_: −3.5, 95% CI: −5.4;−1.6, DG_SA_EDP_: −4.9, 95% CI: −6.8;−3.0) and for esophagus for which a social gradient was found only in sensitivity analyses using England LT (DG_main analyses_: 0.4, 95% CI: −4.2;5.0, DG_SA_Eng LT_: 7.0, 95% CI: 1.4;12.6, DG_SA_EDP_: −0.7, 95% CI: −5.3;3.8).

In women (Figure 1b), clear 5-year deprivation gaps were observed in all three analyses for ENT cancers, bile ducts, esophagus, and cervix uteri. The results showed a trend towards a social gradient in net survival, which was moderate in the three analyses for bladder, corpus uteri, lung, kidney, liver, pancreas, and melanoma, with close values in the 5-year deprivation gaps. Results regarding ovarian cancer, stomach cancer, sarcoma, and CNS cancer were identical according to the three analyses, with negligible 5-year deprivation gaps. The social gradient found in the main analyses was attenuated, but it remained clear with English LT, but not in sensitivity analyses using EDP for breast cancer (DG_main analyses_: 5.1, 95% CI: 3.0;7.3, DG_SA_Eng LT_: 2.5, 95% CI: 0.3;4.6, DG_SA_EDP_: 1.4, 95% CI: −0.7;3.6) and colorectal cancer (DG_main analyses_: 5.4, 95% CI: 2.2;8.6, DG_SA_Eng LT_: 3.5, 95% CI: 0.4;6.7, DG_SA_EDP_: 2.8, 95% CI: −0.4;6.0).

For cancers with a worse prognosis (i.e., lung, liver, pancreas, esophagus, bile ducts, and stomach), the results were stable in all three analyses and for both men and women when considering 1-year survival.

### 3.2. Comparison of Main and Sensitivity Analyses Based on Flexible Modeling

(a)Same model selected in the three analyses.

As presented in Table 1, the same flexible model was selected by the three analyses for 8/14 cancers in men and 11/17 cancers in women, meaning that the effect of EDI on EMH was identical according to the three analyses. Among these consistent cases, results showed that for four cancers in men (bile ducts, CNS, sarcoma, and stomach) and three cancers in women (kidney, lung, and sarcoma), there was no effect of EDI on EMH according to the three analyses. For four cancers in men (ENT, liver, lung, and melanoma) and five cancers in women (breast, CNS, ENT, esophagus, and liver), EDI had an overall proportional effect on EMH in the three analyses according to model selection (Table 1). For these nine cases, although EHR were reduced in both sensitivity analyses (Figure 2), the curves of EHR as a function of EDI according to the three analyses overlapped for liver, esophagus, and CNS cancers in women, or followed the same pattern with close values between main and sensitivity analyses (ENT in men and women, breast in women, lung in men), though sometimes with loss of EHR significance (according to 95% CI) in sensitivity analyses (liver and melanoma in men). The effect of EDI on EMH was found to be time-dependent for cervical and stomach cancer in woman and time- and/or age-dependent for corpus uteri according to the three analyses (Table 1). Curves of EHR of p90 versus p10 of EDI (EHR_p90/p10_) as a function of time (Figure 3) according to the three analyses overlapped for cervical and stomach cancer and followed the same pattern with close values for corpus uteri (with lower EHR_p90/p10_ in sensitivity analyses as compared to main analyses).

(b)Different models selected in the three analyses.

On the other hand, the selected model was different for 6/14 cancers in men and 6/17 cancers in women (Table 1). Among these cases we can distinguish consistent (i), mitigated (ii), inconsistent (iii), and inconsistent and inversed (iv) results.

(i)For cancer of the bile ducts in women (Figure 4), an effect of EDI on EMH was found in all three analyses, which was time-dependent in main analyses (with EHR_p90/p10_ reaching a maximum of 1.96, 95% CI: 1.12;3.43 at 3.8 years of follow-up) and proportional in sensitivity analyses (EHR_p90/p10 (SA_Eng LT)_: 1.38, 95% CI: 1.13;1.69; EHR_p90/p10 (SA_EDP)_: 1.37, 95% CI:1.12;1.68).(ii)Mitigated results were observed for three cancers in men (colon–rectum, kidney, and pancreas) and four in women (colon–rectum, pancreas, melanoma, and ovarian).

For colon–rectum cancers in men and melanoma and ovarian cancers in women, sensitivity analysis using EDP showed no effect of EDI based on model selection as opposed to the two other analyses. Regarding colon–rectum in men (Figure 4), main analyses showed a time-dependent effect of EDI on EMH (EHR_p90/p10_ reaching a maximum of 1.37, 95% CI: 1.12;1.66 at 5 years of follow-up), while sensitivity analysis using English LT presented an overall proportional (according to model selection) but moderate effect (EHR_p90/p10_: 1.05, 95% CI: 0.99;1.12). Regarding melanoma in women (Figure 4), in the main analysis, the effect of EDI on EMH was proportional (according to model selection) and moderate (her_p90/p10_: 1.21, 95% CI: 0.86;1.70), while in the sensitivity analysis using English LT, it was time- and/or age-dependent (according to model selection) but moderate according to the 95% CI. Regarding ovarian cancer (see Appendix A, Appendix A), the effect of EDI was found proportional (according to model selection) according to main and sensitivity analysis using English LT, with similar values of EHR as a function of EDI (curves almost overlapped) but a moderate effect of EDI on EMH according to 95% CI in both analyses.

For colon–rectum cancers in women, kidney cancer in men, and pancreas cancer both in men and women, the three analyses found an overall effect of EDI according to the model selection but with partial loss of significance of this effect (according to 95% CI) during the follow-up in at least one of the three analyses. Regarding colon–rectum cancer in women (Figure 4), very similar results were found according to main and sensitivity analysis using English LT, with an overall proportional effect of EDI on EMH (EHR_p90/p10 (main analyses)_: 1.15, 95% CI: 1.06;1.24; EHR_p90/p10 (SA_Eng LT)_: 1.09, 95% CI: 1.02;1.17), whereas a more mitigated situation occurred in sensitivity analyses using EDP, where the effect of EDI on EMH was observed only at the very beginning of follow-up. A similar situation was observed in women regarding pancreas cancers (Figure 4), with a more moderate effect of EDI on the EMH (EHR_p90/p10 (main analyses)_: 1.08, 95% CI: 0.99;1.17; EHR_p90/p10 (SA_Eng LT)_: 1.07, 95% CI: 0.98;1.16). Regarding pancreas cancer in men, the effect of EDI on EMH was observed around the date of diagnosis but not thereafter (time-dependent effect) in main analysis, and an overall proportional (according to model selection) but moderate effect was found in both sensitivity analyses (EHR_p90/p10 (SA_Eng LT)_: 1.06, 95% CI: 0.99;1.15; EHR_p90/p10 (SA_EDP)_: 1.06, 95% CI: 0.98;1.15). Regarding kidney cancer in men, results were contrasted between main and sensitivity analyses (Figure 4), since main analyses reported a proportional effect of EDI on EMH (EHR_p90/p10_: 1.24, 95% CI: 1.07;1.43), while in both sensitivity analyses a more complex, time-dependent effect was found, with a social gradient at the time of diagnosis, an inverse social gradient at the end of follow-up, and no effect of EDI on EMH in between.

(iii)Regarding bladder (both in men and women) and esophagus cancers in men, an overall proportional effect of EDI on EMH was found in the main analyses according to model selection (with EHR_p90/p10 (bladder, men)_: 1.24, 95% CI: 1.09;1.4; EHR_p90/p10 (bladder, women)_: 1.14, 95% CI: 0.97;1.35; EHR_p90/p10 (esophagus, men)_: 1.13, 95% CI: 1.01;1.26), but no effect was found in any sensitivity analysis for these cancer sites.(iv)Finally, regarding prostate cancer (Figure 5), a time- and/or age-dependent effect of EDI on EMH was found in the main analysis, with an EHR of p90 versus p10 of EDI around 2 for 60- and 70-year-olds, and around 1.5 for 90-year-olds, and no effect at the end of follow-up. In both sensitivity analyses, a time-dependent effect of EDI on EMH was found, with an inverse social gradient reaching a maximum at 5 years of follow-up (EHR_p90/p10 (SA_Eng LT)_: 0.37, 95% CI: 0.19;0.72; EHR_p90/p10 (SA_EDP)_: 0.27, 95% CI: 0.13;0.55).

## 4. Discussion

In summary, in non-parametric analyses, the results were consistent across the three analyses for 9/14 cancers in men and 15/17 cancers in women. They were inconsistent but with similar trends remaining for 3/14 cancers (bladder, colon–rectum, and lung) in men and for 2/17 cancers in women (breast and colon–rectum), and they were fully inconsistent only for 2 cancers in men (prostate and esophagus). According to flexible modelling, the results were consistent for 8/14 cancers in men and 12/17 cancers in women. They were mitigated for 2/14 cancers in men (colon–rectum and pancreas) and 4/17 cancers in women (colon–rectum, pancreas, ovarian, and melanoma), for which the social gradient was attenuated for all or part of the follow-up, or only in one of the sensitivity analyses, and final conclusions were not contradictory. However, the results were inconsistent for 4/14 cancers in men (esophagus, bladder, kidney, and especially prostate) and 1/17 cancer in women (bladder).

As expected, the social gradient in cancer net survival was reduced in sensitivity analyses using simulated deprivation-specific LT, suggesting an overestimation of the social gradient in cancer net survival in main analyses (using the original French LT not stratified in deprivation). This might result from the absence of consideration of the social gradient in general population mortality due to the social determination of other causes of mortality. Overall, our main results were not contradicted, since for most cancers, a significant social gradient in survival remained in sensitivity analyses or at least a tendency was still observed. Nevertheless, this study alerts us to the fact that the overestimation, in few cases, can be so important that conclusions might be questioned, such as those for prostate cancer. The study should therefore warn researchers about this important problem of accounting for the social gradient of mortality in the general population as well as the importance of developing national deprivation-specific LT to produce accurate results in cancer net survival analyses.

While the overall results were consistent or attenuated in sensitivity analyses, for some cancers the results were surprising and not fully understood. For prostate cancer, which is a cancer with very good survival, an explanation could be that the least-deprived patients do not die from other causes as often as those which are the most deprived. Consequently, they are more likely to die from prostate cancer. Conversely, least-deprived patients are more likely to die from other causes and less likely from prostate cancer, resulting in a reversed social gradient. A study in the United States [35] comparing net survival analyses using national vs. state-specific life tables pointed out a possible bias resulting from men with prostate cancer diagnosed at early stages having a better health status than the US general population. We can also make that hypothesis in our own study, and, also assuming that men with prostate cancer diagnosed at early stages are more likely to be less deprived, it could contribute to explaining the inconsistencies found for prostate cancer. The ecologic bias of the EDI could also contribute to this surprising result. Also, one can suppose that the correction brought by these simulated LT is in fact too strong compared to reality (strong overestimation of the expected social gradient of general mortality). Consequently, when the excess mortality is very small (as in the case of prostate cancer) this could lead to false corrected results. Regarding kidney, bladder, or esophageal cancers, the hypotheses to explain such inconsistent results are less obvious but our findings at least suggest that the social gradient in cancer net survival observed in the main analyses might be in a large part explained by the social gradient in background survival, which is also important to consider in the cancer care management and follow-up of patients as part of their general healthcare. We could also assume that socially-stratified risk factors and comorbidities play a part. Surprisingly, previous research [6,36,37,38] relying on appropriate deprivation-specific LT still found a social gradient in cancer net survival for bladder, kidney, esophagus and prostate (with no inverse gradient). One study [39] (using appropriate deprivation-specific LT) did not find either an association between deprivation or cancer net survival for bladder (in women), kidney and esophagus (in men). However, the issues of social inequalities in health and cancer care management may substantially differ across countries, and the statistical methods used in these studies were not always comparable, which could explain such conflicting results. The overestimation of the correction could also explain the discrepancies with some results of previous studies.

Results from the main and sensitivity analyses were even more consistent among women, possibly resulting from lower social inequalities in mortality in the general population than for men [40,41,42].

Although studies of this kind are scarce, similar sensitivity analyses have been conducted [5,15]. In particular, Antunes et al. [15] also found a disappearance of the social gradient of survival for colorectal cancer after the stratification of LT on the socio-economic level in Portugal, but no other cancer site was studied. Ito et al. [5] also found a systematic reduction in deprivation gaps when using simulated deprivation-specific LT for Japan in sensitivity analyses, but they did not found any inversed social gradient or inconsistent results as compared to us. However, this study was based only on a non-parametric method. Our research brings additional information about the impact of using inappropriate LT in cancer net survival analyses, and it relies on a wide variety of cancers, and large-sample, high-quality population-based data as well as very robust statistical analyses.

One strength of this study was the conduction of comparisons between the three analyses using both a non-parametric method and flexible modelling. This allowed us to obtain a confirmation of the results through two different and validated methods, including a very sophisticated one (flexible modelling of excess mortality hazard using multidimensional penalized splines) which permitted adjustments and the consideration of possible non-linear and non-proportional effects as well as interactions [29,30]. The results from the two methods were consistent, both showing inconsistent results for esophageal and prostate cancer in men. Flexible modelling additionally highlighted the overestimation of the social gradient in the main analyses for kidney cancer in men and bladder cancer in both men and women.

Even though this work allowed us to point out possible cancer sites for which the social gradient in cancer survival was overestimated in our previous study, it is important to note that, the present analyses being based on simulated and not real data, we were not able to precisely quantify the potential bias resulting from the lack of stratification on deprivation of the original French LT nor to shed light on the true extent of the overestimation. Moreover, social inequalities in the background mortality from external sources and thus in the simulated LT were not based on the same variable as in the net survival analyses since the index used in England LT was not the EDI but the IMD (i.e., aggregated ecological index at the contextual level) and the indicator used in EDP was the income (i.e., single indicator at the individual level). In the EDP, the net income per consumption unit of a given household as well as the education level and socio-professional categories were available. All three variables are known to be good indicators of socio-economic position. We considered net income per consumption unit of the household to be the most relevant for use in our analyses, because it represents a continuum and can be classified into five categories according to its natural distribution in the population without the need to create arbitrary categories. Although EDI and IMD are different in terms of construction (e.g., geographic scale) and information provided, they are probably more comparable to each other than income. Indeed, using an individual measure of deprivation such as income to stratify the LT was not the most appropriate way in a study based on an ecological aggregated index such as EDI. The correction of the LT by the income indicator was probably too extreme to study deprivation with EDI. This would explain why the results from the sensitivity analyses using English LT were generally intermediate between results from main analyses (strongest social gradient) and sensitivity analyses using EDP (smallest social gradient). On the other hand, while English LT provided a social gradient of the background mortality based on a quite close index as compared to EDI, this cannot be considered as a perfect reflection of the social gradient in background mortality in France, since the social inequalities in mortality are not the same between these two countries [16]. EDP data were the only source providing information on the social gradient in background mortality for the French population. It therefore seemed interesting to also consider French data, so we chose to use both external sources and produced two sets of simulated LT. Additionally, using deprivation-specific LT from a European country with substantial social inequalities in background mortality (such as Lithuania, according to Mackenbach et al. [16]) would have allowed us to consider a “worst case scenario” for these sensitivity analyses. We also acknowledge some limitations regarding the implementation of the simulated LT. In the set simulated using English LT, we applied the social gradient of mortality of England, but this also changes over time (between 2006 and 2013, duplicating the data from 2011 in the years 2012 and 2013, as described in the Methods section and in the Appendix A) without knowing whether this trend was comparable to that of France, whereas in the LT simulated using EDP data, we assumed that the social gradient of mortality from the period 2012–2016 was stable over time in the French general population.

Regarding other possible weaknesses of the study, we cannot be certain of the absence of noise for some results due to the lack of power after stratifying the analyses by cancer site and by sex. Moreover, data on the stage at cancer diagnosis were not available, and as discussed in Tron et al. 2019 [11], this lack of information limits the interpretation of the results regarding the social gradient in cancer net survival. However, this is unlikely to have a substantial impact on the results of the sensitivity analyses per se. 

Given all of these elements, results from neither the main nor the sensitivity analyses can be considered as right or “true” results. These can only be determined when validated deprivation-specific French LT are available, which should be a future research and development prospect, as it has been done recently in Portugal [43]. Also, the problem investigated in this paper only occurs when the estimation of net survival is based on excess mortality rates, and it does not concern studies relying on the causes of death (which have their own limitations).

In conclusion, except for prostate cancer and to a lesser extent bladder (among both men and women), esophageal (among men), and kidney (among men) cancers, this study confirms previously published findings about the disparities in cancer net survival according to the socioeconomic environment for several cancer sites in France, despite the absence of consideration given to the social gradient in background mortality in the LT, which was a main limitation to this previous work. Nevertheless, the present study indicates how important it is to create proper deprivation-specific LT in France in order to accurately investigate the social gradient in cancer net survival, which should be done in the next coming years.

## Figures and Tables

**Figure 1 cancers-15-00659-f001:**
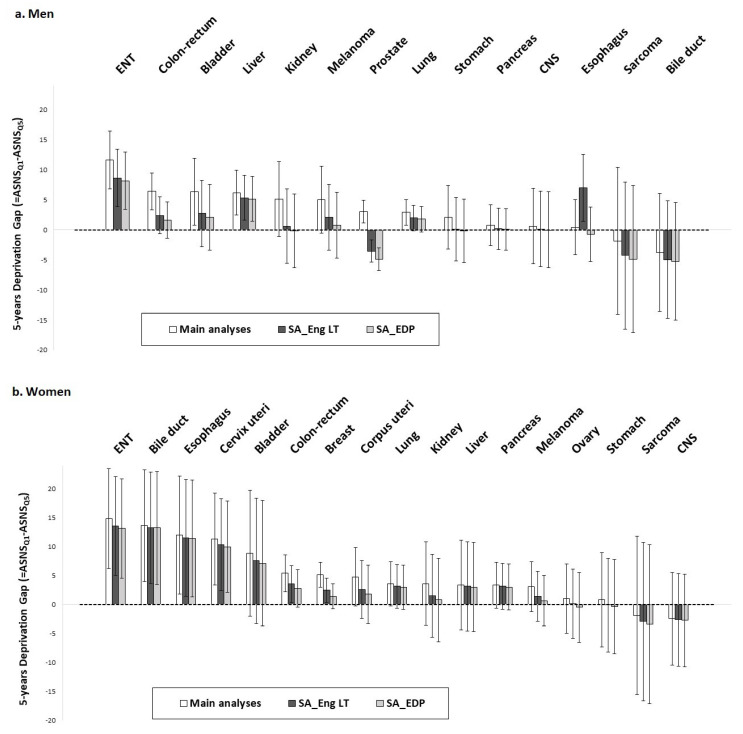
Difference in 5-year age-standardized net survival (ASNS) between patients living in least-deprived environment (Q1) and patients living in most-deprived environment (Q5) (i.e., 5-Year Deprivation Gap), estimated by non-parametric Pohar–Perme method, for men (**a**) and women (**b**), for each cancer site, according to main analyses, sensitivity analyses using English life tables (SA_Eng LT), and sensitivity analyses using EDP data (SA_EDP). ASNS: age-standardized net survival; CNS: central nervous system; EDI: European Deprivation Index; EDP: French permanent demographic sample [Echantillon Démographique Permanent]; ENT: ear, neck, throat (head and neck cancers); Q1: 1st quintile of the national distribution of EDI; Q5: 5th quintile of the national distribution of EDI; SA_EDP: sensitivity analyses based on EDP data; SA_Eng LT: sensitivity analyses based on English life tables.

**Figure 2 cancers-15-00659-f002:**
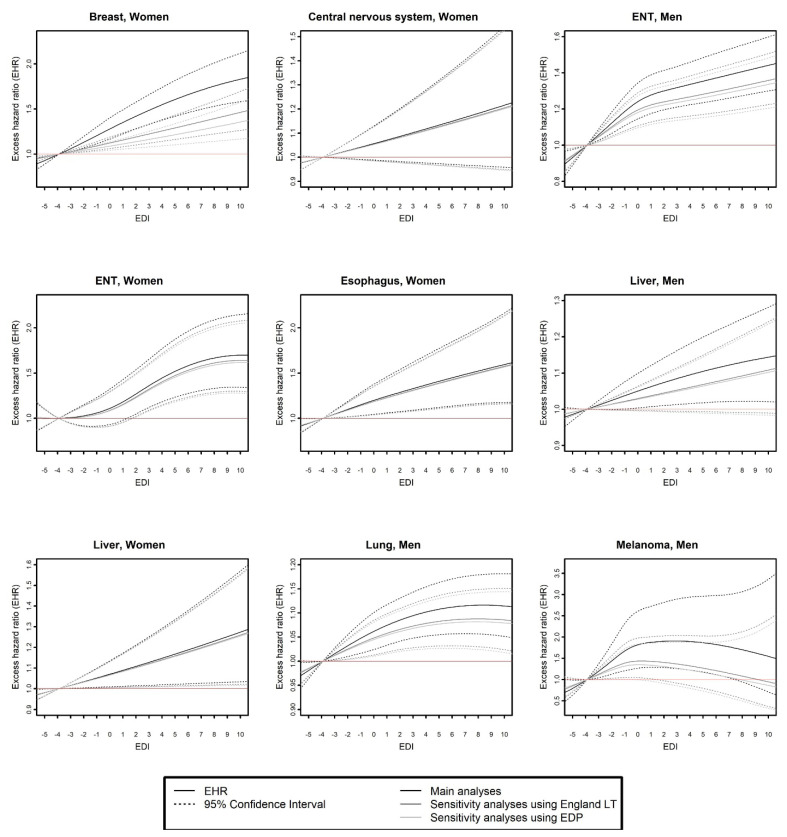
Excess mortality hazard ratio (EHR) as a function of deprivation (EDI) according to main analyses, sensitivity analyses using English life tables, and sensitivity analyses using EDP data, for cancer sites for which all three analyses were concordant (same flexible model selected), and model selection showed an overall proportional effect of EDI on excess mortality hazard (independent of age and time of follow-up). EDI: European Deprivation Index; EDP: French permanent demographic sample [Echantillon Démographique Permanent]; EHR: excess mortality hazard ratio; ENT: ear, neck, throat (head and neck cancers); LT: life tables.

**Figure 3 cancers-15-00659-f003:**
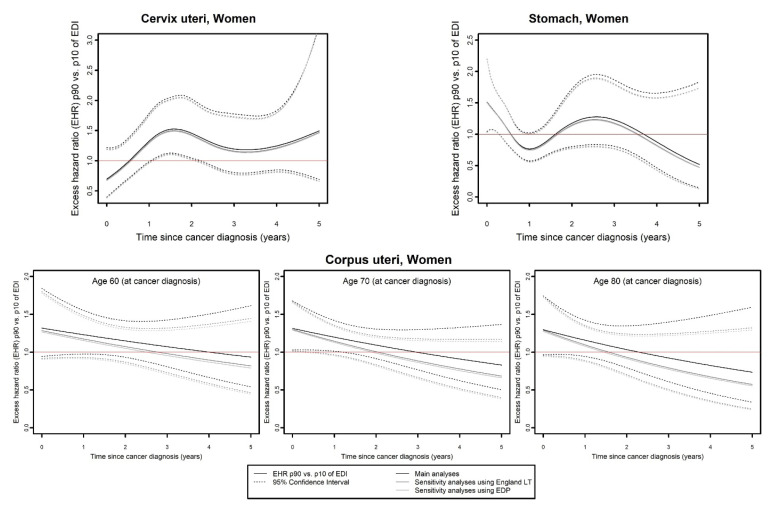
Excess mortality hazard ratio (EHR) between the 90th and the 10th percentile of the national distribution of EDI as a function of the time since cancer diagnosis, according to main analyses, sensitivity analyses using English life tables and sensitivity analyses using EDP data, for cancer sites for which all three analyses were concordant (same flexible model selected) and model showed an overall time-dependent effect of EDI on excess mortality hazard. For corpus uteri cancer, the effect of EDI on excess mortality was also age-dependent, therefore the results are presented for three values of age corresponding to the 25th, 50th, and 75th percentiles of the age distribution in the studied population with corpus uteri cancer. EDI: European Deprivation Index; EDP: French permanent demographic sample [Echantillon Démographique Permanent]; EHR: excess mortality hazard ratio; LT: life tables; p90, p10: 90th and 10th percentile of the national distribution of EDI.

**Figure 4 cancers-15-00659-f004:**
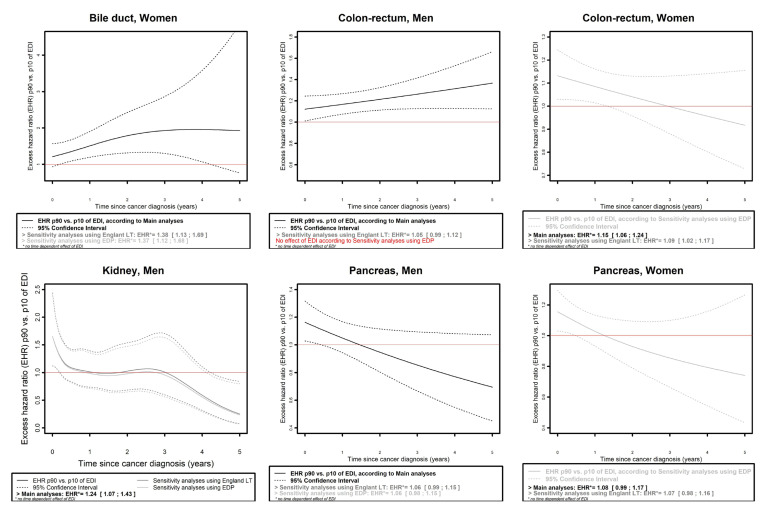
Excess mortality hazard ratio (EHR) between the 90th and the 10th percentile of the national distribution of EDI, as a function of the time since cancer diagnosis according to main and sensitivity analyses, for cancer sites for which the effect of EDI was different according to the analysis (i.e., different models selected in the three analyses) and time-dependent. For melanoma in women, the effect of EDI on excess mortality was also age-dependent in the sensitivity analysis using English life tables, therefore results are presented for 3 values of age corresponding to the 25th, 50th and 75th percentiles of the age distribution in the studied female population with melanoma. EDI: European Deprivation Index; EDP: French permanent demographic sample [Echantillon Démographique Permanent]; EHR: excess mortality hazard ratio; LT: life tables; p90, p10: 90th and 10th percentile of the national distribution of EDI.

**Figure 5 cancers-15-00659-f005:**
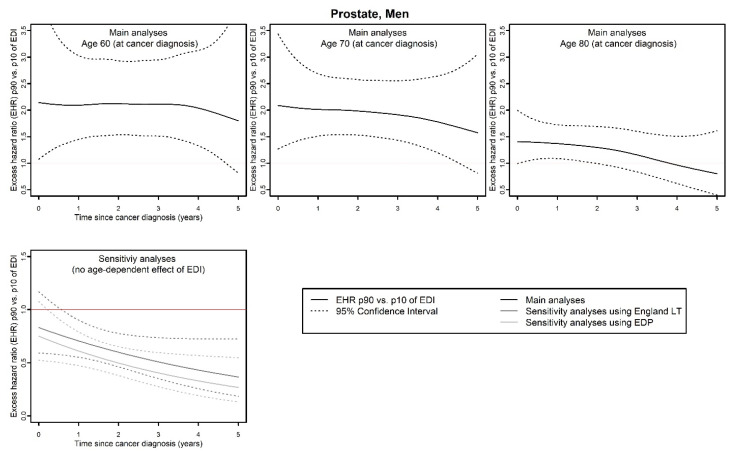
Excess mortality hazard ratio (EHR) between the 90th and the 10th percentile of the national distribution of EDI, as a function of time since cancer diagnosis and according to the main and sensitivity analyses, for prostate cancer (discordant results between main and sensitivity analyses). EDI: European Deprivation Index; EDP: French permanent demographic sample [Echantillon Démographique Permanent]; EHR: excess mortality hazard ratio; LT: life tables; p90, p10: 90th and 10th percentile of the national distribution of EDI.

**Table 1 cancers-15-00659-t001:** The effect of deprivation (EDI) on cancer net survival for each cancer site, from the selected flexible model, according to main analyses, sensitivity analyses using English life tables (SA_Eng LT), and sensitivity analyses using EDP data (SA_EDP).

	Men			Women		
	Main Analyses	SA_Eng LT	SA_EDP	Main Analyses	SA_Eng LT	SA_EDP
Bile duct	No effect	No effect	No effect	Time-dependent	Proportional	Proportional
Bladder	Proportional	No effect	No effect	Proportional	No effect	No effect
Breast	-	-	-	Proportional	Proportional	Proportional
Cervix uteri	-	-	-	Time-dependent	Time-dependent	Time-dependent
CNS	No effect	No effect	No effect	Proportional	Proportional	Proportional
Colon-rectum	S and time-dependent	Proportional	No effect	Proportional	Proportional	Time-dependent
Corpus uteri	-	-	-	Time- and/or age-dependent	Time- and/or age-dependent	Time- and/or age-dependent
ENT	Proportional	Proportional	Proportional	Proportional	Proportional	Proportional
Esophagus	Proportional	No effect	No effect	Proportional	Proportional	Proportional
Kidney	Proportional	Time-dependent	Time-dependent	No effect	No effect	No effect
Liver	Proportional	Proportional	Proportional	Proportional	Proportional	Proportional
Lung	Proportional	Proportional	Proportional	No effect	No effect	No effect
Melanoma	Proportional	Proportional	Proportional	Proportional	Time- and/or age-dependent	No effect
Ovary	-	-	-	Proportional	Proportional	No effect
Pancreas	Time-dependent	Proportional	Proportional	Proportional	Proportional	Time-dependent
Prostate	Time- and/or age-dependent	Time-dependent	Time-dependent	-	-	-
Sarcoma	No effect	No effect	No effect	No effect	No effect	No effect
Stomach	No effect	No effect	No effect	Time-dependent	Time-dependent	Time-dependent

CNS: central nervous system; EDI: European Deprivation Index; EDP: French permanent demographic sample [Echantillon Démographique Permanent]; ENT: ear, neck, throat (head and neck cancers); SA_EDP: sensitivity analyses based on EDP data; SA_Eng LT: sensitivity analyses based on English life tables.

## Data Availability

Data and code are available upon reasonable request.

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
