# Peer review of "Is the Social Gradient in Net Survival Observed in France the Result of Inequalities in Cancer-Specific Mortality or Inequalities in General Mortality?"

_cancers, 2023, doi:10.3390/cancers15030659_

Round 1

Reviewer 1 Report

Dear Author(s),

your manuscript entitled “To what extent social deprivation gap in cancer net survival observed in France could be attributable to cancer-specific or general mortality” investigates the effect of socially determined comorbidities on the social gradient observed in net survival for patients diagnosed with cancer and alerts the scientific community about the possible bias due to the use of general life tables rather than deprivation-specific life tables. 

Although this is a relevant topic, since social inequalities in health are a major concern in European countries and evidence on social gradient in cancer survival is fundamental to address health care policies and to reduce cancer burden, the paper presents some weaknesses and needs a revision.

There is one major point regarding the aim of the study: in the introduction (see page 3, lines 132-133) as well as in the abstract (see page 2, lines 61-62) you declare that the aim of the manuscript is “to evaluate to what extent the social gradient in cancer net survival observed in France could be attributable to inaccurate LT” , also the title recalls this aim. However, in the summary of the manuscript (see pages 1-2, lines 53-55) as well as in the discussion (see page 14-15 lines 474-479, and page 15, lines 508-511) you also state that: “the present analyses being based on simulated and not real data, we were not able to precisely quantify the potential bias resulting from the lack of stratification on deprivation of the original French LT nor to put light on the true extent of the overestimation”. I suggest to reformulate the aim in order to address this inconsistency.

Minor points are listed in the following:

1)    According to what you write in the discussion one weakness of the study is that social inequalities in background mortality from external sources and thus in the simulated life tables were not based on the same variable as in the net survival. I wonder if you have explored the possibility of using education as socio-economic variable, which is easier to measure in a comparable way across different data sources and in the adult population is related to general mortality and to cancer survival. I suggest you to add at least a comment on this point;

2)    It is well known that stage at diagnosis is a very strong predictor of short- and medium-term survival, it is also well known that there are inequalities in the access to cancer screening that might cause a different stage distribution according to the socioeconomic status. In the present study, the stage at diagnosis has not been taken into account, it is probably not available in the dataset used, however a comment on it should be included in the discussion;

3)    In the analysis you use 5-years net survival as reference indicator, however for worse prognosis cancer types considered in the analyses, like for example lung, pancreas, stomach and esophagus, looking at 1-year net survival in addition to 5-years would be informative and relevant for the study purposes;

4)    The interpretation about inconsistencies found out for prostate cancer is not very convincing I suggest you to also consider the effect of overdiagnosis of tumors not clinically relevant which are identified by means of PSA opportunistic screening. This effect is possibly different according to the socioeconomic status of the patient. I suggest you to have a look at the results from Stroup et al published on the Journal of National Cancer Institute in 2014.

Reviewer 2 Report

Major Positives

The paper addresses a real issue and important issue. A comprehensive set of cancer sites are tested. Two alternative ways to simulate LT by deprivation are provided for greater robustness. Great attention is paid to the analytic methods, non-parametric & parametric. The authors are transparent about the shortcomings of the study. The study does provide some evidence in the round that their research question/concern is valid, in that where results differed across a number of sites the deprivation-stratified LTs generally attenuated the gradient in net survival.

Major Negatives

Although the authors state that a large dataset has been used, the width of confidence intervals suggest that the study is under-powered for the majority of the cancer sites reported. It is hard to place confidence on particular site results as both Type 1 and Type 2 errors likely to be prevalent. These makes the reading of the results section laborious, and interpretation of the [selected] graphs difficult, the credibility of ‘best’ parametric models in doubt. The authors do a lot of work in the discussion, but hard to shake the suspicion that much of the more-detailed interpretation could be building off noise. Is the lack of consistency in attenuation change in gradient by cancer site due mainly to noise or are there good ‘signal’ reasons for it (different deprivation distribution, different relative sizes of excess to expected mortality, counteracting gradient of excess rates), even for the big cancer sites? I am guessing that the use of deprivation-stratified LT has been justified in other countries, and therefore this paper needs to justify its use in France, and so the low-power is a weakness.

It is hard to visualise what the internal and external deprivation categories represent, and whether they would capture the deprivation gradient in the same way, or if one is superior. I understand that there is a fundamental difficulty/issue here of the latent deprivation variable and how that would be defined and best captured.

Given that the particular results are difficult to interpret, and that the deprivation categories hard to visualise, I wonder could the general question have been demonstrated in a simpler way by reviewing the typical relative rates for deprivation quintiles observed in other countries, and adjusting the French National LT by a typically low, medium, high difference in RR between quintiles and describing the impact. Were the RR’s for the external sources given in the paper? Were these realistic in the EDP? Were they comparable to England?

Round 2

Reviewer 2 Report

I am happy that my questions have been adequately dealt with. Perhaps, some of the answer to external sources could be made more explicit in the methods (there are discussed later in the discussion) strengthening their rationale and complementarity earlier on?